# CLIPDRAG: COMBINING TEXT-BASED AND DRAG-BASED INSTRUCTIONS FOR IMAGE EDITING

**Ziqi Jiang, Zhen Wang, Long Chen**[†]
The Hong Kong University of Science and Technology
{zjiangbl, zwangjr}@connect.ust.hk, longchen@ust.hk

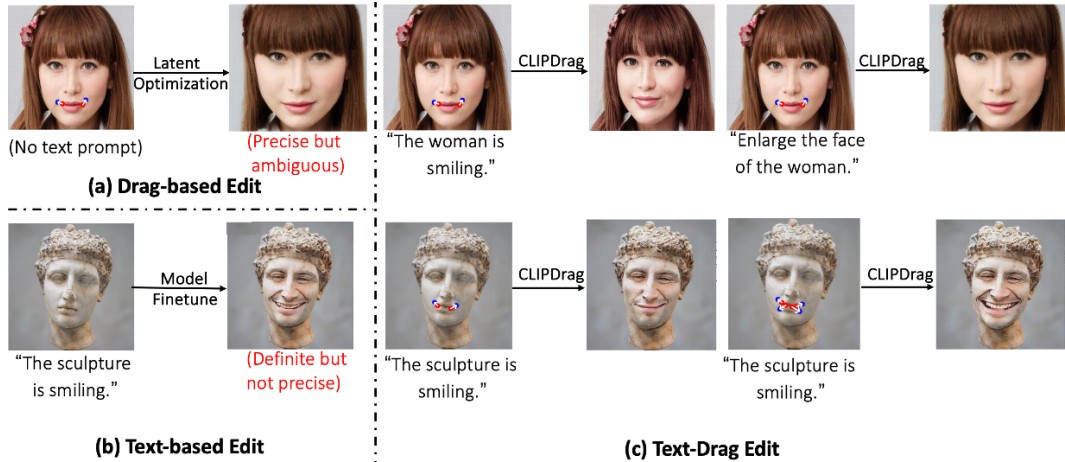

Figure 1: Three different kinds of image editing methods. (a): Drag-based Editing. Users need to click handle points (red) and target points (blue). (b): Text-based Editing. Only the edit prompt is needed to perform the edit. (c): Text-Drag Editing. both drag points and edit prompts are required.

## ABSTRACT

Precise and flexible image editing remains a fundamental challenge in computer vision. Based on the modified areas, most editing methods can be divided into two main types: global editing and local editing. In this paper, we discussed two representative approaches of each type (*i.e.*, text-based editing and drag-based editing. Specifically, we argue that both two directions have their inherent drawbacks: Text-based methods often fail to describe the desired modifications precisely, while drag-based methods suffer from ambiguity. To address these issues, we proposed **CLIPDrag**, a novel image editing method that is the first try to combine text and drag signals for precise and ambiguity-free manipulations on diffusion models. To fully leverage these two signals, we treat text signals as global guidance and drag points as local information. Then we introduce a novel global-local motion supervision method to integrate text signals into existing drag-based methods (Shi et al., 2024b) by adapting a pre-trained language-vision model like CLIP (Radford et al., 2021). Furthermore, we also address the problem of slow convergence in CLIPDrag by presenting a fast point-tracking method that enforces drag points moving toward correct directions. Extensive experiments demonstrate that CLIPDrag outperforms existing single drag-based methods or text-based methods.

## 1 INTRODUCTION

Recently, notable breakthroughs in diffusion models (Ho et al., 2020; Song et al., 2020a;b), have led to many impressive applications (Meng et al., 2021; Dong et al., 2023; Kumari et al., 2023). Among

---

[†]Long Chen is the corresponding author. Codes: https://github.com/ZiQi-Jiang/CLIPDrag.

them, image editing is recognized as a significant area of innovation and has gained enormous attention (Kim et al., 2022; Nichol et al., 2021; Sheynin et al., 2024; Valevski et al., 2023). Generally, the goal of this task is to edit realistic images based on various editing instructions. Mainstream methods for image editing can be coarsely categorized into two groups: 1) **Global Editing**, edits given images by a text prompt (Li et al., 2023; Kawar et al., 2023) or an extra image (Zhang et al., 2023a; Epstein et al., 2023) containing global information of the desired modification. Most of them involve finetuning a pre-trained diffusion model. 2) **Local Editing**, mainly consists of drag-based methods (Pan et al., 2023). This framework requires users to click several handle points (*handles*) and target points (*targets*) on an image, then perform the semantic editing to move the content of handles to corresponding targets. Typical methods (Mou et al., 2023; Shi et al., 2024b) usually contain a motion supervision phase that progressively transfers the features of handles to targets by updating the DDIM inversion latent, and a point tracking phase to track the position of handles by performing a nearest search on neighbor candidate points.

Although the aforementioned methods have gained significant prominence, two drawbacks of them cannot be overlooked. **1) Imprecise Description in Global Editing**. Global editing such as text-based image methods (Kawar et al., 2023) is unambiguous but is hard to provide detailed edit instructions. For example, in Figure 1(b), the prompt (*i.e.*"The sculpture is smiling") tells the model how to edit the image, but it is difficult to provide fine-grained editing information, like the extent of the smile. **2) Ambiguity Issues in Local Editing.** Although local editing like drag-based editing methods can perform precise pixel-level spatial control, they suffer from ambiguity because the same handles and targets can correspond to multiple potential edited results. For example, in Figure 1(a), there exist two edited results meeting the drag requirements: one enlarging the face, the other making the woman smile. A natural approach to solving the ambiguity problem is to add more point pairs, but it does not work in real practice. This is because the diffusion network is a Markov chain and related errors accumulate as the update continues. Thus when adding more points, it usually means more update iterations, which will result in the degradation in image fidelity. In conclusion, local editing is precise but ambiguous while global editing is exactly the opposite.

Since these two kinds of editing are complementary, it is natural to ask: can we combine these two control signals to guide the image editing process? In this way, text signals can serve as global information to reduce ambiguity. Meanwhile, drag signals can act as local control signals, providing more detailed control. However, combining these two signals presents two challenges: 1) **How to integrate two different kinds of signals efficiently?** This is difficult because text-based and drag-based methods have completely different training strategies. Specifically, most text-based methods (Kim et al., 2022) require finetuning a pre-trained diffusion model to gradually inject the prompt information. But drag-based methods (Shi et al., 2024b) typically involve freezing the diffusion model and only optimizing the DDIM inversion latent of the given image. Besides, the update in text-based editing often involves the whole denoising timesteps while drag-based approaches only focus on a specific timestep. 2) **How to solve the optimization problem and maintain the image quality?** Previous drag-based methods are very slow in some situations. Sometimes the handles will stuck at one position for many iterations. In worse situations, they even move in the opposite direction of targets. This phenomenon becomes more serious when adding text signals because combining two different signals usually requires more optimization steps. Thus we need a better approach to optimize the update process while maintaining fidelity as much as possible.

To address these problems, we propose CLIPDrag, the first method to combine text-based and drag-based approaches to achieve precise and flexible image editing (For brevity, we denote this kind of editing as *text-drag edit*). This approach was built based on the general drag-based diffusion editing framework, which optimizes the DDIM inversion latent of the original image at one specific timestep. Specifically, we propose two modules to solve the aforementioned problems after a typical identity finetuning process. 1) **Global-Local Motion Supervision (GLMS)**. The key of GLMS is to utilize the gradient from text and drag signals together for ambiguity-elimination. Specifically, for text information, we obtain the global gradient by backward of the global CLIP loss for the latent. Similarly, for drag points, the local gradient is calculated using a similar paradigm in DragDiffuion (Shi et al., 2024b). Because CLIP guidance methods cannot operate at a single fixed step, simply adding these two gradients is ineffective. Specifically, in GLMS, We disentangle the global gradient into two components: *Identity Component* perpendicular to the local gradient to maintain the image's global structure information, and *Edit Component* parallel to the local gradient, which will be combined with the local gradient from drag signals to edit the image. By comparing their

direction, we choose different gradient fusion strategies to either perform the edit or maintain the image identity. 2) **Fast Point Tracking (FPT)**. As we mentioned before, combining drag and text signals will result in slow convergence in the latent optimization. Thus we propose a faster tracking method to accelerate the optimization process. FPT is introduced to update the position of handles after each GLMS operation. Specifically, when performing the nearest neighbor search algorithm, FPT masks all candidates that are far away from current targets. Consequently, handles will become closer to targets after each update, and FPT also ensures that the moving trajectory of handles will not be repetitive. It is observed that FPT significantly accelerates the image editing process and improves the image quality to some extent.

Combining these good practices, CLIPDrag achieves high-quality results for text-drag editing. Extensive experiments demonstrate the effectiveness of CLIPDrag, outperforming the state-of-the-art approaches both quantitatively and qualitatively. To summarize, our contributions are as follows:

- We have pointed out that previous drag-based and text-based image editing methods have the problems of ambiguity and inaccuracy, respectively.

- We propose CLIPDrag, a solution that incorporates text signals into drag-based methods by using text signals as global information.

- Extensive experiments demonstrate the superiority and stability of CLIPDrag in text-drag image editing, marking a significant advancement in the field of flexible and precise image editing.

## 2 RELATED WORK

**Diffusion Models.** Diffusion models are a class of generative models that progressively degrade data by adding noise and then learn the reverse denoising process to generate realistic data. The Denoising Diffusion Probabilistic Model (DDPM) (Ho et al., 2020) was the first to demonstrate that diffusion models could achieve results comparable to state-of-the-art GANs (Goodfellow et al., 2020) for unconditional image generation. Subsequent works, such as DDIM (Song et al., 2020a) and ScoreSDE (Song et al., 2020b), refined the theoretical framework and improved performance. Classifier-guided (Dhariwal & Nichol, 2021) and classifier-free guidance (Ho & Salimans, 2022) methods further explored the potential of diffusion models for conditional generation. Inspired by the impact of large language models, the computer vision community has also trained diffusion models on large datasets to achieve commercial-grade performance, leading to the development of models such as GLIDE (Nichol et al., 2021), EMU (Sheynin et al., 2024), and Imagic (Kawar et al., 2023). Among these, Stable Diffusion (SD) (Rombach et al., 2022) has become particularly popular due to its efficiency. Unlike previous methods, SD projects images into a lower-dimensional latent space before adding noise, significantly reducing memory and computational requirements. Building on SD, numerous downstream applications have been proposed, including personalization (Ruiz et al., 2023), style transfer (Zhang et al., 2023b), and inpainting (Lugmayr et al., 2022).

**Text-based Image Editing.** Unlike text-to-image generation which involves creating an image from scratch (Ho et al., 2022; Dhariwal & Nichol, 2021; Saharia et al., 2022; Gu et al., 2022), text-based image editing is about altering certain areas of a given image. DiffCLIP (Kim et al., 2022) leverages contrastive language-image pertaining (CLIP) (Radford et al., 2021) to fine-tune the diffusion process, enhancing diffusion models for high-quality zero-shot image editing. SINE (Zhang et al., 2023c) improves editing performance on single images by utilizing a large-scale pre-trained text-to-image model. Paint by Example (Yang et al., 2023) is the first approach to employ self-supervised training for more precise control through exemplar guidance. FlexiEdit (Koo et al., 2024) enhances non-rigid editing by reducing the high-frequency components in the target areas. Prompt-to-Prompt (Hertz et al., 2022) refines text-based editing by manipulating cross-attention maps during diffusion, enabling more effective modifications. Similarly, MasaCtrl (Cao et al., 2023) achieves complex non-rigid image editing by converting self-attention into mutual attention. InstructPix2Pix (Brooks et al., 2023) integrates a pre-trained large language model with a text-to-image model to generate training data for a conditional diffusion model, facilitating direct image editing from textual prompts. Null-text inversion (Mokady et al., 2023), on the other hand, enhances editing by optimizing default null-text embeddings to achieve desired transformations. While these text-based methods empower users to edit images using natural language, they often lack the precision and explicit control offered by drag-based editing techniques.

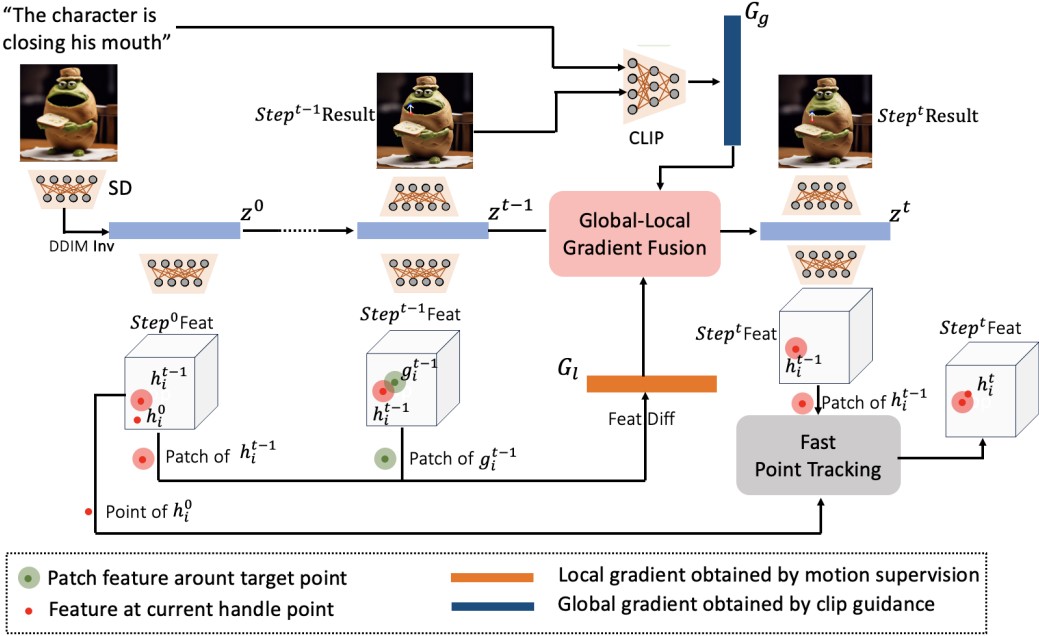

Figure 2: Illustration of our scheme for an intermediate single-step optimization. $z^t$ means the optimized latent code at $t^{th}$ updation. The local gradient and global gradient are calculated by backwarding the motion supervision loss and CLIP guidance loss with respect to the latent code respectively. Then the Global-Local Gradient Fusion method is introduced to combine these two information to update the latent code. The new handles are inferred through our fast point tracking method.

**Drag-based Image Editing.** Recently, DragGAN (Pan et al., 2023) introduced a novel method for achieving precise spatial control over specific regions of an image based on user-provided drag instructions. Building on it, DragDiffusion (Shi et al., 2024b) applied the approach to diffusion models, achieving superior performance and establishing a benchmark for future work. FreeDrag (Ling et al., 2023) reduces the burden of point tracking by introducing two key components: adaptive updates of template features and a line search with the backtracking mechanism. StableDrag (Cui et al., 2024) proposes a discriminative point-tracking method to construct a stable drag-based editing framework. SDE-Drag (Meng et al., 2021) offers a straightforward yet powerful approach for point-based content manipulation based on the stochastic differential equation (SDE) framework. LightningDrag (Shi et al., 2024a) enables high-quality image editing by eliminating the need for time-consuming latent optimization. Similarly, InstantDrag (Shin et al., 2024) achieves a fast editing framework by incorporating two optical flow generators. A concurrent work, RegionDrag (Lu et al., 2024), addresses ambiguity issues by transforming the problem into region-based editing through a gradient-free copy-paste operation. Unlike our method, RegionDrag employs a simpler transfer mechanism for achieving edits. Another line of drag-based methods includes DragonDiffusion (Mou et al., 2023) and DiffEditor (Mou et al., 2024), which reformulate the image editing task into a gradient-based process by defining an energy function that aligns with the desired edit results.

## 3 APPROACH

**Problem Formulation.** Given an image $I$ and two text prompts $P_o$ and $P_e$, where $P_o$ describes the original image and $P_e$ depicts the edited image. After choosing $n$ handles $\{h_1, h_2, \ldots, h_n\}$ and corresponding targets $\{g_1, g_2, \ldots, g_n\}$, a qualified text-drag editing method has to satisfy the following two requirements: 1) Move the feature of point $h_i$ to $g_i$, while preserving the irrelevant content of the original image. 2) The edited image must align with the edit prompt $P_e$.

**General Framework**: As shown in Figure 2[1], our method CLIPDrag, consists of three steps:

---

[1]While we use "SD" in the diagram, it may only refer to part of the diffusion model. For example, the "DDIM Inv" only utilizes the VAE component.

1) Identity-Preserving Finetuning (Sec 3.1): Given the input image $I$ and its description $P_o$[1], We first finetune a pre-trained latent diffusion model $\epsilon_\theta$ using the technique of low-rank adpation (LoRA (Hu et al., 2021)). After the finetuning, we encode the image into the latent space and obtain the latent $z_t$ at a specific timestep $t$ through DDIM inversion. $z_t$ will be optimized for many iterations to achieve the desired edit, and each iteration consists of two following phases.

2) Global-Local Motion Supervision (Sec 3.2): We denote the latent at $t$-th step and handle point $p_i$ during the $k$-th iteration as $z_t^k$ and $p_i^k$ respectively. At $k$-th iteration, we can calculate global and local gradient vectors for $z_t^k$. These two gradients will be combined by our Global-Local Gradient Fusion method, and the final result is used to update $z_t^k$ to $z_t^{k+1}$.

3) Fast Point Tracking (Sec 3.3): After GLMS, we can update the position of handles with the nearest neighbor search. To accelerate the editing speed while preserving the image fidelity, we propose FPT to ensure that new handles are closer to their corresponding targets by filtering all candidates that are far away from the targets.

## 3.1 IDENTITY-PRESERVING FINETUING

As analyzed in previous work (Shi et al., 2024b), directly optimizing the latent $z_t$ in diffusion-based methods will cause the problem of image fidelity. So finetuning on a pre-trained diffusion model is necessary to encode the features of the image into the U-Net. Specifically, the image and its description prompt $P_o$ are used to finetune the diffusion model $\epsilon_\theta$ through the LoRA method:

$$\mathcal{L}_{ft}(z, \Delta\theta) = \mathbb{E}_{\epsilon,t}[\|\epsilon - \epsilon_{\theta+\Delta\theta}(\alpha_t z + \sigma_t \epsilon)\|_2^2] \tag{1}$$

where $z$ is the latent space feature map concerning image $I$. $\theta$ and $\Delta\theta$ represent the U-Net (Ronneberger et al., 2015) and LoRA parameters, $\alpha_t$ and $\sigma_t$ are constants pre-defined in the diffusion schedule, $\epsilon$ is random noise sampled from distribution $\mathcal{N}(0, I)$. After the finetuning, we choose a specific timestep $t$ and obtain the DDIM inversion latent (Song et al., 2020a) as follows:

$$z_{t+1} = \sqrt{\alpha_{t+1}}(\frac{z_t - \sqrt{1-\alpha_t} \cdot \epsilon_\theta(z_t)}{\sqrt{\alpha_t}}) + \sqrt{1-\alpha_{t+1}} \cdot \epsilon_\theta(z_t) \tag{2}$$

the latent will be optimized in the subsequent process while keeping all other parameters frozen.

## 3.2 GLOBAL-LOCAL MOTION SUPERVISION

This step aims to combine text and drag signals, as shown in Figure 3. We will introduce how to process each control information, and then show how to combine them.

**CLIP-guidance Gradient**. We extract knowledge from text signal using a local direction CLIP loss (Kim et al., 2022), which aligns the direction between the source image $I$ and generates image $\hat{I}$ with the direction between original prompt $P_o$ and edit prompt $P_e$:

$$\mathcal{L}_{direction}(I, \hat{I}, P_o, P_e) := 1 - \frac{<\Delta I, \Delta T>}{\|\Delta I\|\|\Delta T\|} \tag{3}$$

where $\Delta I = E_I(\hat{I}(z_t^k)) - E_I(I)$, $\Delta T = E_T(P_e) - E_T(P_o)$. Here $E_T$, $E_I$ represents text and image encoder from a pre-trained CLIP model. However, the identity component of $P_e$ is canceled out during the calculation of $\Delta T$, making it difficult to maintain the image identity. Besides, sometimes it is impossible to calculate the direction loss because $P_o$ is not pro-

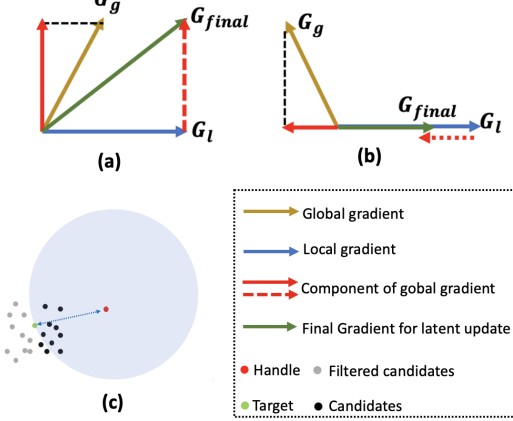

Figure 3: (a) $G_g$ is consistent with $G_l$. (b) $G_g$ contradicts with $G_l$. (c) Fast Point Tracking.

---

[1] For convenience, users can opt not to provide a description, in which case a captioning model, such as GPT-4V will automatically generate a prompt.

vided (Zhang et al., 2024). Consequently, we choose the global target loss to extract more information from edit prompt $P_e$ as follows:

$$\mathcal{L}_{global}(\hat{I}, P_e) := 1 - \frac{< E_I(\hat{I}(z_t^k)), E_T(P_e) >}{\|E_I(\hat{I}(z_t^k))\|\|E_T(P_e)\|} \quad (4)$$

Then we can obtain the global gradient ($G_g$) from the text signal: $G_g = \partial\mathcal{L}_{global}/\partial z_t^k$. Later we will explain how to use $G_g$ to maintain the image identity and guide the edit.

**Drag-guidance Gradient**. We denote the U-Net output feature maps obtained by $k$-th updated latent $z_t^k$ as $F(z_t^k)$. And the gradient from drag signals is obtained by the motion supervision loss $L_{ms}$, which is the difference between the features of corresponding targets and handles:

$$\mathcal{L}_{ms}(z_t^k) = \sum_{i=1}^{n} \sum_{q\in\Omega(p_i,r1)} \|F_{q+d_i}(z_t^k) - sg(F_q(z_t^0))\|_1 + \|(z_t^k - sg(z_t^0)) \odot (\mathbb{1} - M)\|_1 \quad (5)$$

Where $\Omega(p_i, r1) = \{(x, y) : \|x - x_i\| \leq r_1, \|y - y_i\| \leq r_1\}$, $M$ is an optional mask, and $r_1$ represents the patch radius. $sg(\cdot)$ represents the stop gradient operation (van, 2017), $di = (g_i - h_i^k)/\|g_i - h_i^k\|$, is an unit vector from $h_i^k$ to $g_i$. Thus the local gradient ($G_l$) can be calculated by: $G_l = \partial\mathcal{L}_{ms}/\partial z_t^k$.

**Global-Local Gradient Fusion**. Now we explain how to incorporate the two gradients in detail. The key motivation of this method is to decompose the process of text-based editing process into two parts: the edit component and the identity component. Specifically, as shown in Figure 3, when the direction of the edit component from $G_g$ is consistent with $G_l$ (Figure 3(a)), it means both signals agree with how to update the latent. Thus we use $G_l$ to guide the edit while preserving the structure of the image by the identity component of $G_g$. When the two edit directions are contradictory (Figure 3(b)), we choose to correct the drag direction using the editing component, inspired by (Zhu et al., 2023). This approach can be formalized as follows:

$$G_{final} = \begin{cases} G_l + \lambda\sin\langle G_g, G_l\rangle \cdot G_g, & \cos\langle G_g, G_l\rangle > 0, \\ G_l - \lambda\cos\langle G_g, G_l\rangle \cdot G_g, & \cos\langle G_g, G_l\rangle < 0, \end{cases} \quad (6)$$

where $G_{final}$ means the final gradient to update the latent code and $\lambda$ is a hyper-parameter.

## 3.3 FAST POINT TRACKING

Although CLIP guidance can relieve the ambiguity problem, it makes the optimization of GLMS more difficult. In drag-based methods, a similar optimization issue arises when more point pairs are added. In the previous point tracking strategy, handles sometimes get stuck in one position or move far away from their corresponding targets. This significantly slows down the editing process. To remedy this issue, we add a simple constraint on the point tracking process: when updating the handles through the nearest neighbor search algorithm, only consider the candidate points that are closer to the targets, as shown in Figure 3(c). Our FPT method can be formulated as follows:

$$h_i^{k+1} = \underset{a\in\Omega(h_i^k, r2)\ \&\ dis(a, g_i) < dis(h_i^k, g_i)}{\arg\min} \|F_q(z_t^{k+1}) - F_{h_i^k}(z_t^0)\| \quad (7)$$

where $dis(a, g_i)$ represents the distance between point $a$ and target $g_i$, $r_2$ is the search radius.

## 4 EXPERIMENTS

### 4.1 IMPLEMENTATION DETAILS

We used Stable Diffusion 1.5 (Rombach et al., 2022) and CLIP-ViT-B/16 (Dosovitskiy et al., 2020) as the base model. For the LoRA finetuning stage, we set the training steps as 80, and the rank as 16 with a small learning rate of 0.0005. In the DDIM inversion, we set the inversion strength to 0.7 and the total denoising steps to 50. In the Motion supervision, we had a large maximum optimization step of 2000, ensuring handles could reach the targets. The features were extracted from the last layer of the U-Net. The radius for motion supervision ($r_1$) and point tracking ($r_2$) were set to 4 and 12, respectively. The weight $\lambda$ in the Global-Local Gradient Fusion process was 0.7.

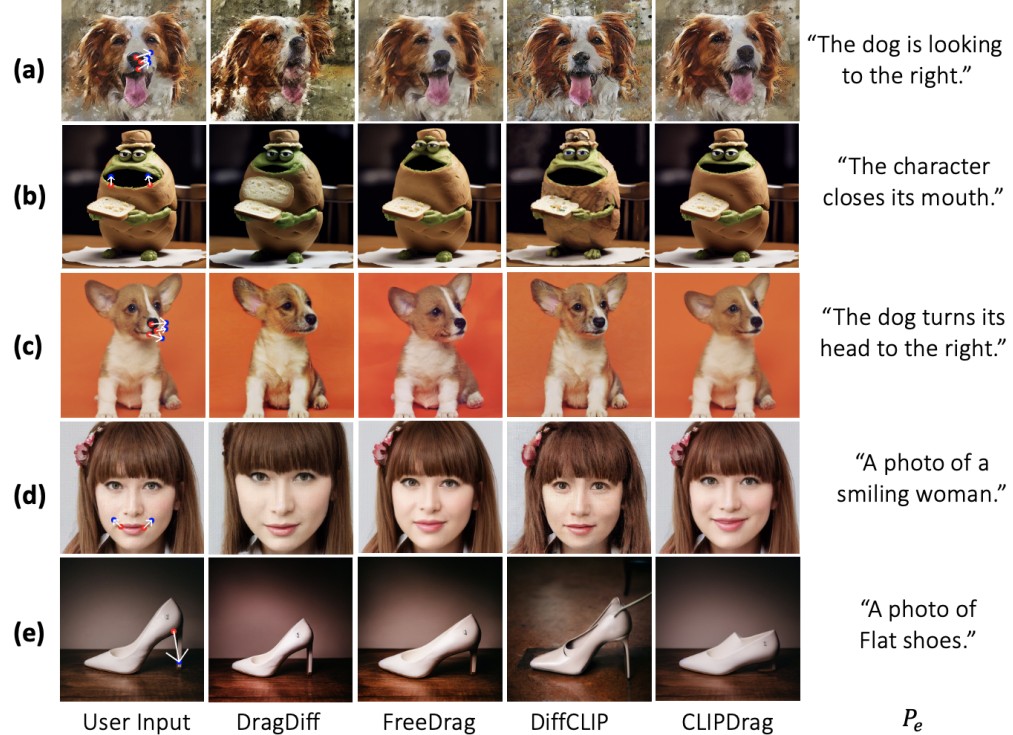

Figure 4: Comparisons with Drag-based methods (DragDiff, FreeDrag) and Text-based methods (DiffCLIP). $P_e$ is the edited prompts, which are required by CLIPDrag and DiffCLIP.

## 4.2 TEXT-DRAG EDITING RESULTS

**Settings**. To show the performance of CLIPDrag we compared both drag-based methods (DragDiffusion, FreeDrag, RegionDrag, StableDrag, InstantDrag, LightningDrag), and text-based method (DiffCLIP) on text-drag image editing tasks. Specifically, drag-based methods require drag points as edit instructions while text-based methods need an edit prompt to perform the modification. For CLIPDrag, both editing prompts and drag points are required to perform the edit. All input images are from the DRAGBENCH datasets (Shi et al., 2024b).

**Results**. As illustrated in Figure 4, our method shows better performance over the two different editing frameworks. Due to the limited space, the results of RegionDrag, StableDrag, InstantDrag, and LightningDrag are shown in Appendix A. Compared with text-based methods (DiffCLIP), CLIPDrag can perform more precise editing control on the pixel level. Compared with drag-based methods (DragDiffusion, FreeDrag), CLIPDrag successfully alleviates the ambiguity problem, as shown in Figure 4(b)(d)(e). This is because former drag-based methods intend to perform structural editing like moving or reshaping, instead of semantic editing such as emotional expression modification. It is reasonable because moving an object is much easier than changing its morphological characteristics. Consequently, these models prefer to choose the shortcut to realize feature alignment in motion supervision, resulting in the ambiguity problem. Our proposed method effectively solves the issue, by introducing CLIP guidance as the global information to point out a correct optimization path.

Besides, former drag-based approaches cannot guarantee edited image quality when multiple drag point pairs exist. So we also give some examples with multiple point pairs to compare the stability of these methods. As shown in Figure 4(a)(c), these results validate the effectiveness of the two techniques in our method: identity component's guidance to preserve the image quality, and fast point tracking to achieve better drag performance.

More results of CLIPDrag are shown in Figure 5. The leftmost three examples verify the effectiveness of our method: by combining the information of drag points and edit prompts ($P_e$), CLIPDrag achieves an edit with high image fidelity and no ambiguity. The middle three examples show the

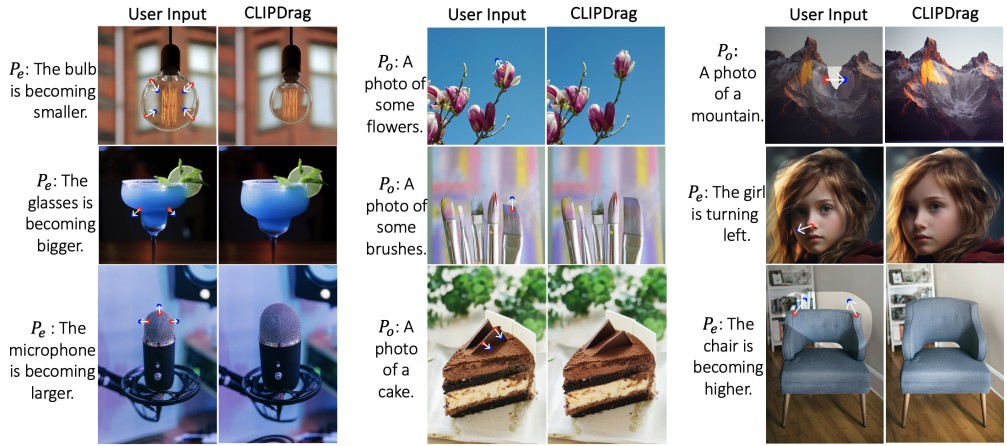

Figure 5: Some examples of CLIPDrag. For each input, both an edit prompt and drag points are required. $P_e$ and $P_o$ represent the edit prompts and original prompts respectively.

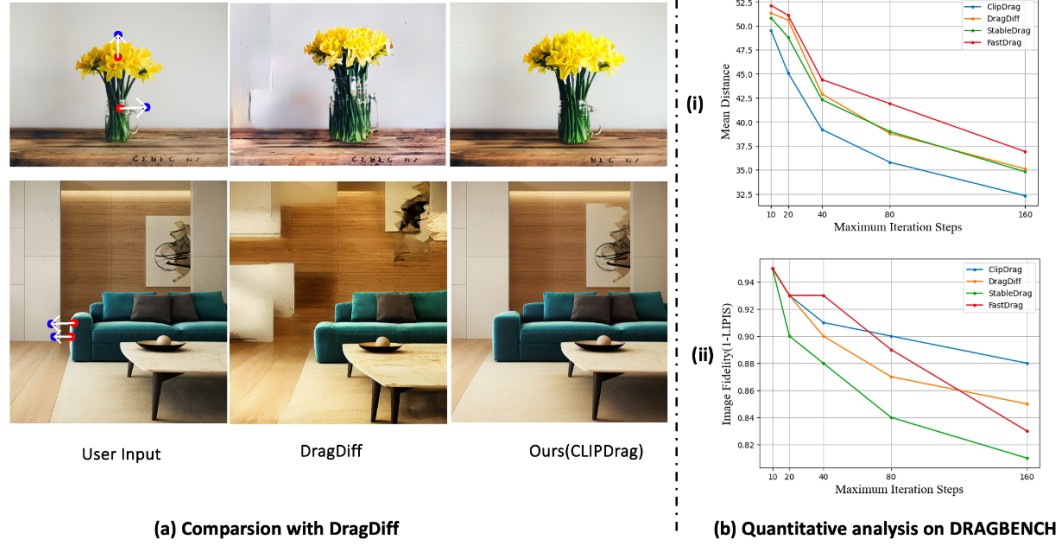

Figure 6: (a): Examples of CLIPDrag and DragDiff on drag-based edit. Since no text information is needed in the editing process of DragDiff, we use the finetuning prompt ($P_o$) as the editing prompt ($P_e$) in CLIPDrag for fairness. (b): The quantitative experiment on the DRAGBENCH dataset.

situations when users do not want to consider the ambiguity or find it difficult to describe the desired edit. And we found CLIP also works well when the edit prompt is replaced with the original prompt ($P_o$). The rightmost three examples show the results when adding a mask.

## 4.3 DRAG-BASED EDITING RESULTS

**Settings**. Since our method is based on general drag-based frameworks, we also explored the performance of CLIPDrag in pure drag-based editing tasks. Specifically, we replaced the editing prompt ($P_e$) with the corresponding original prompt ($P_o$). This ensures that no extra text information will be introduced. We compared our CLIPDrag with DragDiffusion, FastDrag, and StableDrag on the DRAGBENCH benchmark with five different max iteration step settings. To evaluate the image fidelity, we reported the average 1-LIPIS score (IF). Besides, Mean distance (MD) was calculated to show the distance between the final handles and targets (Lower MD represents better drag performance). Due to the limited space, the visual results of StableDrag and FastDrag are included in Appendix E.

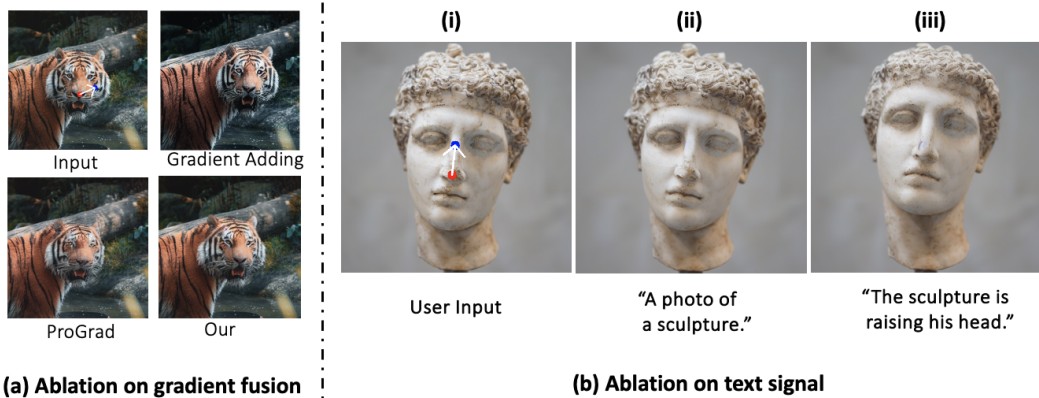

Figure 7: **(a)**: Ablation study on different gradient fusion strategies. "Gradient Adding" represents simply adding the global and local gradient, and "ProGrad" is the method proposed in (Zhu et al., 2023). The difference between ProGrad and ours is that we utilize the identity component to maintain image fidelity. **(b)**: Ablation on text signals. for the same drag instruction (*e.g.*(ii)), different results are obtained with different edit prompts ((ii) & (iii)).

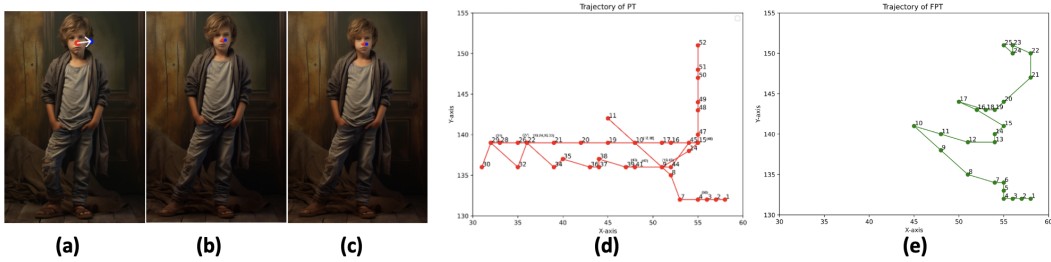

Figure 8: Ablation study on two kinds of point tracking strategies: original point tracking in DragDiff (PT) and our fast point tracking (FPT). for the same drag instruction (a), (b)(c) are the results of PT and FPT, respectively. (d)(e) are the corresponding trajectories of the handle point.

**Results**. Quantitative results are shown in Figure 6(b). In this figure, the $x$-axis represents the max iteration steps and the $y$-axis represents the IF and MD metrics respectively. As can be seen, CLIPDrag has better performance than DragDiff, FastDrag, and StableDrag. Specifically, on most max iteration steps settings, CLIPDrag has higher IF and lower MD. This means handle points are closer to the targets in CLIPDrag, in the meanwhile, the edited image quality is preserved or even improved (Figure 6(a)).

## 4.4 ABLATION STUDY

**Ablation on Text Signals**. We performed ablation studies to clarify the effect of text signals in CLIPDrag by using different edit prompts and keeping the drag points unchanged.

*Results.* As illustrated in Figure 7(b), the ambiguity can be alleviated in CLIPDrag. For example, the drag instruction (i) corresponds to at least two different edit directions: one moving up the sculpture (ii), the other raising his head (iii). By giving different text signals we can choose different edit paths. We also observed that when no extra text information is given, CLIPDrag tends to align handle and target features by simple position translation (ii), instead of semantic edit (iii).

**Ablation on the Global-Local Gradient Fusion**. We studied the effect of different methods to combine text and drag signals. We ran this ablation experiment with three different strategies global-local gradient fusion: adding two gradients together, ProGrad, and our method.

*Results.* As shown in Figure 7(a), adding two gradients cannot alleviate the ambiguity problem because the CLIP guidance does not take effect on the single effect, as illustrated in DiffCLIP. The ProGrad method can relieve the ambiguity problem but performs badly in maintaining the quality

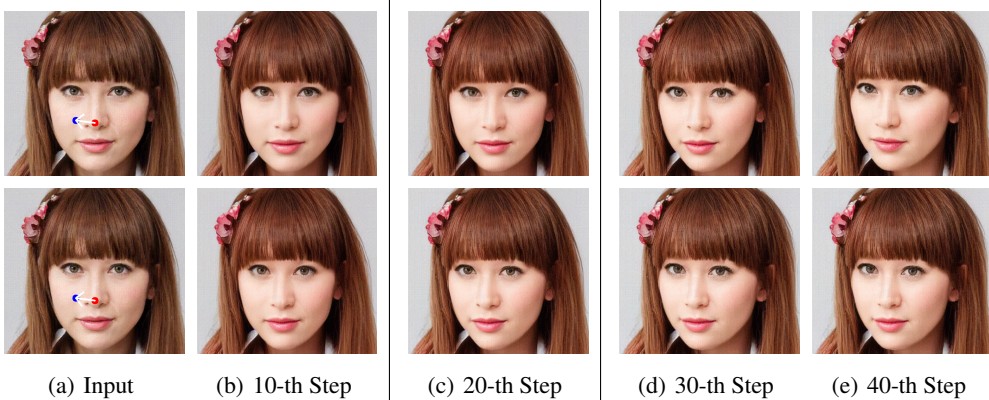

|          |              |              |              |              |
|:--------:|:------------:|:------------:|:------------:|:------------:|
| (a) Input | (b) 10-th Step | (c) 20-th Step | (d) 30-th Step | (e) 40-th Step |

Figure 9: Further analysis on Point Tracking strategies. For the same input, intermediate edit results are visualized at multiple optimization steps.

of edited images. This is because it ignores the identity component in text signals, which contains information about the image's structure. By contrast, our proposed method can effectively fuse the two signals, not only relieving the ambiguity but also maintaining the image fidelity to some degree.

**Ablation on Different Point Tracking Strategies**. Finally, We showed the effect of different point tracking strategies. We based this experiment on our CLIPDrag method while replacing the fast point tracking with the normal one in DragDiff (Shi et al., 2024b). Besides, to make the result more convincing, we tried to make edited images from these two methods similar by adjusting the random seed and learning rate, while keeping other parameters like patch radius unchanged.

*Results.* As shown in Figure 8, when achieving similar editing results, FPT effectively reduces the optimization iterations consumed. From the moving trajectory in Figure 8(d), we found that handles could get stuck at one point or move in circles in previous methods. Instead, handles move closer and closer to the targets in our FPT strategy, thus speeding up the editing process. Figure 8(c) shows that the FPT method does not harm the image quality. To further show the effect of FPT, we give another example and their intermediate results, shown in Figure 9(c). At the 20th iteration, the FPT strategy began to perform semantic editing, while the PT method was still in the stage of identity-preserving.

## 5 CONCLUSION

In this work, we tackled the ongoing challenge of achieving precise and flexible image editing within the field of computer vision. We observed that existing text-based methods often lack the precision for specific modifications, while drag-based techniques are prone to ambiguity. To overcome these challenges, we introduced CLIPDrag, a pioneering method that uniquely integrates text and drag signals to enable accurate and unambiguous manipulations. We enhanced existing drag-based methods by treating text features as global guidance and drag points as local cues. Our novel global-local gradient fusion method further optimized the editing process during motion supervision. Additionally, to address the issue of slow convergence in CLIPDrag, we developed an FPT method that efficiently guides handle points toward their target positions. Extensive experiments clearly demonstrated that CLIPDrag significantly outperforms existing methods that rely solely on drag or text-based inputs.

## ACKNOWLEDGMENT

This work was supported by RGC Early Career Scheme (26208924), National Natural Science Foundation of China Young Scholar Fund (62402408), Huawei Gift Fund, and HKUST Sports Science and Technology Research Grant (SSTRG24EG04).

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

## APPENDIX

The appendix is organized as follows:

1. In Sec. A, we show comparisons with more drag-based methods on text-drag edit.
2. In Sec. B, we show four more examples in the Figure 1.

3. In Sec. C, we show results when we apply text and drag guidance sequentially

4. In Sec. D, we show examples when text and drag signals conflict.

5. In Sec. E, we show examples of StableDrag, FreeDrag on drag-based edit.

6. In Sec. F, we show the inference time comparisons.

## A  MORE RESULTS OF TEXT-DRAG EDIT.

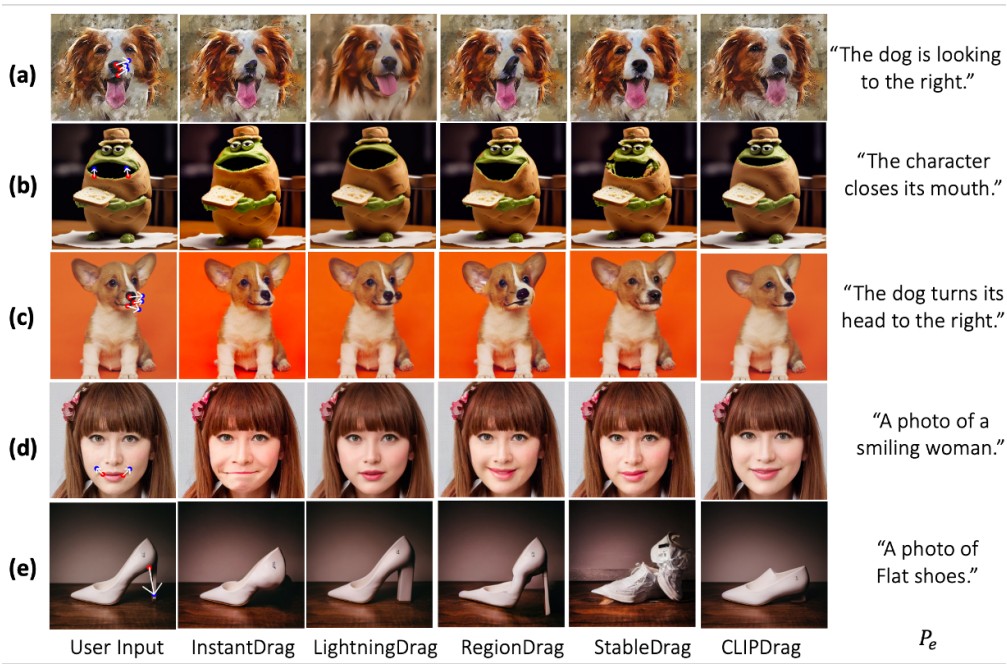

Figure 10: Results of four drag-based methods, including InstantDrag, LightningDrag, RegionDrag, StableDrag.

To better show the superiority of CLIPDrag compared to traditional drag-based methods, we show the results of four other drag-based methods(StableDrag, LightningDrag, RegionDrag, InstantDrag), as shown in Figure 10.

Compared with these methods, CLIPDrag successfully alleviates the ambiguity problem and better maintains the identity , as shown in Figure 10. This is because former drag-based methods intend to perform structural editing like moving or reshaping, instead of semantic editing such as emotional expression modification. It is reasonable because moving an object is much easier than changing its morphological characteristics. Consequently, these models prefer to choose the shortcut to realize feature alignment in motion supervision, resulting in the ambiguity problem. Our proposed method effectively solves the issue, by introducing CLIP guidance as the global information to point out a correct optimization path.

## B  EXPLANATION OF MOTIVATION

To demonstrate our motivation more clearly, we added four results for the second example in Figure 1. Specifically, we provide examples with and without drag edit the following examples. "The sculpture is smiling and not showing his teeth." and "The sculpture is smiling and not raising his head", as shown in Figure 11.

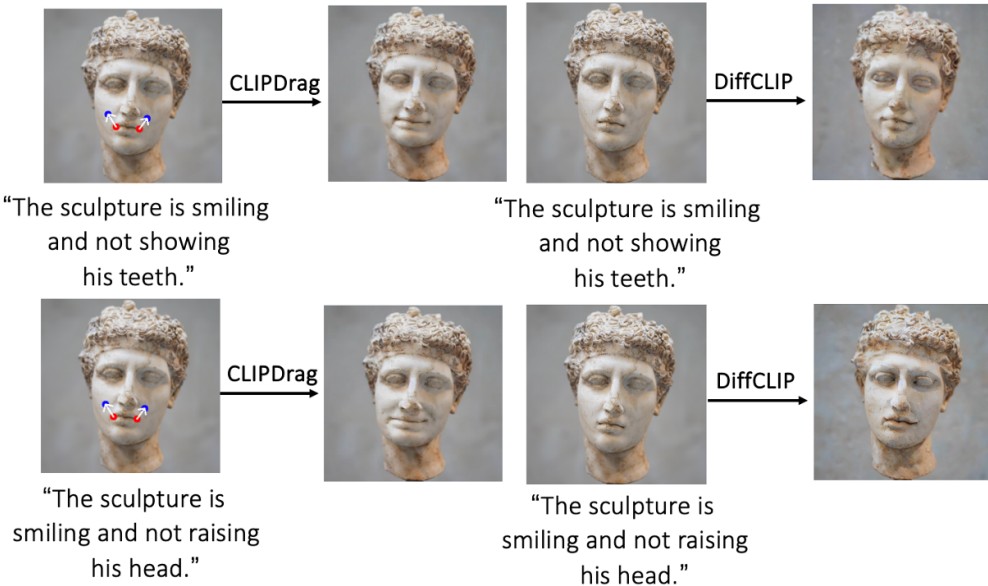

Figure 11: Results of two text prompts with and without drag edit.

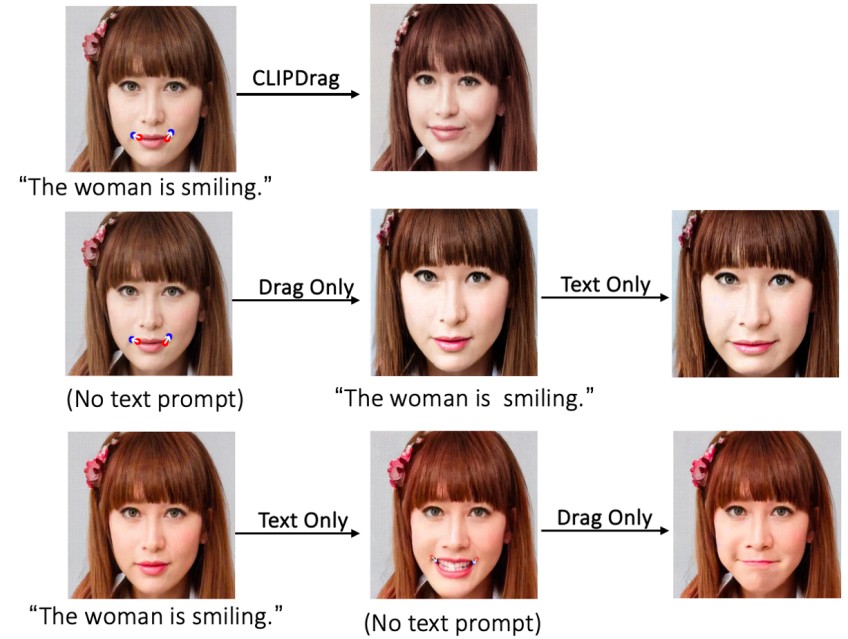

Figure 12: Results when the text guidance and drag guidance are applied sequentially.

## C  APPLY GUIDANCE SEQUENTIALLY

As shown in Figure 12, we demonstrate why applying the two types of guidance sequentially is not considered in our paper. Actually, if we were to apply drag-based editing first, the optimization direction of the latent could be incorrect, meaning that the ambiguity problem would occur before the text guidance is applied. Additionally, if text-based editing were applied after the drag operation, the position of the target points would be altered. As a result, after the text-based edit, the final positions of the handles would no longer align with the targets, which contradicts the core principle of drag-based methods. Instead, when text-based editing is applied first, the position of the handle points is altered. This alteration can mislead the subsequent drag operation.

## D Conflicting Instructions

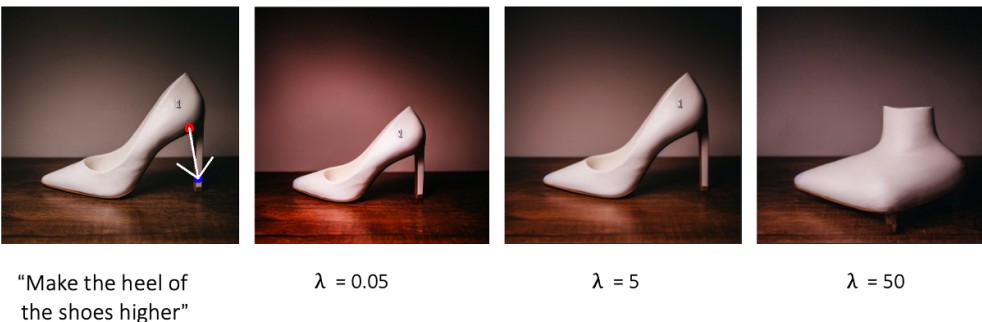

    "Make the heel of         λ = 0.05         λ = 5         λ = 50
    the shoes higher"

Figure 13: Results when the text signals contradict with drag points. $\lambda$ is a hyper-parameter in Equation (6) to control the strength of text guidance.

While our primary motivation is to use the text prompt to complement the local drag edit—ensuring that these two signals are consistent in most cases—we also explored scenarios where these guidance signals conflict. Based on our experiments shown in Figure 13, we observed two types of potential editing outcomes:

**Ambiguity or Neutralization at Moderate Text Strength ($\lambda \leq \mathbf{10}$).** When the strength of the text signal is moderate, ambiguity can arise if the text guidance fails to provide accurate information. In other cases, the text signal may counteract the drag operation, effectively neutralizing its effect. For instance, the drag instruction combined with the prompt "Make the heel of the shoes higher" might yield a result akin to "Make the heel not so high."

**Implausible Results at High Text Strength($\lambda = 100$).** When the text guidance is excessively strong, it overwhelms the denoising process, making it difficult to handle the perturbation. This can result in implausible or unrealistic edits.

## E More results of drag-based edit.

**Settings**. Since our method is based on general drag-based frameworks, we explored the performance of CLIPDrag in pure drag-based editing tasks. Specifically, we replaced the editing prompt ($P_e$) with the corresponding original prompt ($P_o$). This ensures that no extra text information will be introduced. We compared our CLIPDrag with DragDifusion, FastDrag and StableDrag on the DRAGBENCH benchmark with five different max iteration step settings. To evaluate the image fidelity, we reported the average 1-LIPIS score (IF). Besides, Mean distance (MD) was calculated to show the distance between the final handles and targets (Lower MD represents better drag performance).

**Results**. As can be seen in Table (1)(2), CLIPDrag has better performance than DragDiff, FastDrag and StableDrag(the number(10,20,40,80,160) means the maximum iterations). Specifically, on most max iteration steps settings, CLIPDrag has higher IF and lower MD. This means handle points are closer to the targets in CLIPDrag, in the meanwhile, the edited image quality is preserved or even improved, as shown in Figure 14.

| Method | 10 | 20 | 40 | 80 | 160 |
|--------|------|------|------|------|------|
| **Ours** | 49.5 | 45.1 | 39.2 | 35.8 | 32.3 |
| **DragDiff** | 51.3 | 50.6 | 42.9 | 38.8 | 35.1 |
| **StableDrag** | 50.8 | 48.8 | 42.3 | 39.0 | 34.8 |
| **FastDrag** | 52.1 | 51.1 | 44.4 | 41.9 | 36.9 |

Table 1: Mean Distance results.

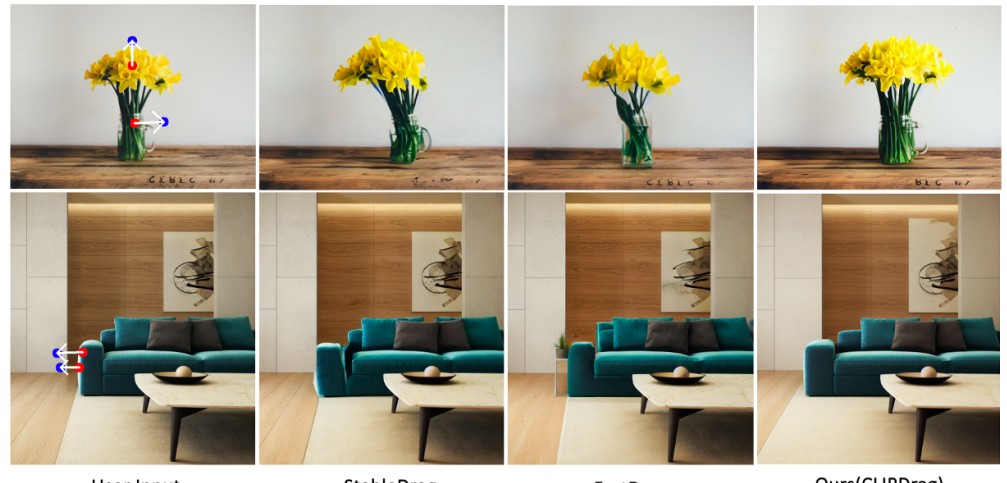

Figure 14: Results of Stable and FastDrag on pure drag-based edit.

| Method | 10 | 20 | 40 | 80 | 160 |
|---|---|---|---|---|---|
| **Ours** | 0.95 | 0.94 | 0.93 | 0.90 | 0.88 |
| **DragDiff** | 0.95 | 0.93 | 0.90 | 0.87 | 0.85 |
| **StableDrag** | 0.95 | 0.90 | 0.88 | 0.84 | 0.81 |
| **FastDrag** | 0.95 | 0.93 | 0.93 | 0.89 | 0.83 |

Table 2: Image Fedlity results.

| Method | DragDiff | FreeDrag | FastDrag | SDE-Drag | StableDrag | CLIPDrag(Ours) |
|---|---|---|---|---|---|---|
| **Time** | 80.3s | 69.4s | 75.5s | 70.0s | 72.3s | 47.8s |

Table 3: Comparisons of inference time.

# F  INFERENCE TIME COMPARISONS

In this section, we report the average inference time of CLIPDrag, DragDiff, SDE-Drag and Free-Drag(the result is calculated on a single 3090 GPU by averaging over 100 examples sampled from the DragBench. As shown in the Table 3, our method is significantly faster than previous works. This is because the inference time is directly correlated with the number of optimization iterations. In CLIPDrag, the text guidance helps to indicate the correct optimization direction, while our FPT strategy prevents handles from moving in the wrong direction or forming loops. Both of these factors reduce the number of iterations required to move the handle points to their target positions, resulting in faster editing speeds.

