# OpenReview forum: "CLIPDrag: Combining Text-based and Drag-based Instructions for Image Editing"
_ICLR.cc/2025/Conference — ICLR 2025 Poster_

### Official Review · Reviewer_CxEc · 2024-10-17

**Soundness:** 3
**Presentation:** 3
**Contribution:** 3
**Rating:** 8
**Confidence:** 5

**Summary:**

This manuscript introduces CLIPDrag, a novel method that integrates text-based and drag-based signals for image editing, leveraging both for precise control and reduced ambiguity. The method utilizes Global-Local Motion Supervision (GLMS) and Fast Point Tracking (FPT) to enhance the image editing process, aiming to outperform existing methods by combining the strengths of both editing approaches.

**Strengths:**

1. **Innovative Integration**: The paper presents a compelling approach by combining text and drag inputs to guide image editing. This dual-input strategy addresses the limitations of each method when used independently, potentially offering more controlled and precise edits.
2. **Technical Depth**: The introduction of GLMS shows a deep understanding of the challenges in image editing, particularly in handling the complexities associated with combining different types of editing signals.
3. **Experimental Validation**: Extensive experiments, including ablation studies, demonstrate the effectiveness of CLIPDrag against state-of-the-art methods. The results are well-presented and support the claims of improved performance in terms of both precision and ambiguity resolution.

**Weaknesses:**

1. **Novelty of FPT**: The paper should acknowledge that searching for handle points along the path from handle points to targets has been previously explored in methods like DragGAN and DragDiffusion. To clarify the unique contributions of FPT, the authors should provide side-by-side comparisons of point searching strategies, highlighting any improvements or distinctions in their approach.

2. **Comprehensive Comparisons**: While the paper compares CLIPDrag with some existing methods, it would benefit from more extensive comparisons or discussions with recent techniques such as InstantDrag, LightningDrag, StableDrag, and RegionDrag. Although these methods may use different training approaches or inputs, incorporating their text-supervision signals could demonstrate CLIPDrag's ability to address ambiguities present in these methods, showcasing its generalizability. Additionally, these methods should be thoroughly discussed in the related work section to provide a more complete context.

3. **Performance Metrics**: The paper should include a discussion or report on inference time comparisons. This information is crucial for understanding the practical applicability of CLIPDrag in real-world scenarios and how it compares to other methods in terms of computational efficiency.

4. **User Input Optimization**: While the text prompt is provided in DragBench, it's worth noting that the original DragGAN paper did not require text input. The additional text prompt in CLIPDrag may increase user effort. To address this, the authors could explore incorporating vision-language models like GPT-4V to automatically interpret the input image (as shown in the first column of Figure 4). This approach could significantly reduce user burden while maintaining the benefits of text-guided editing.

**Questions:**

In the context of drag-based editing where maintaining the original identity of objects while manipulating specific features is a major challenge, your manuscript suggests the integration of text-based inputs to guide the editing process. Could you elaborate on how the addition of text signals specifically contributes to preserving the object's original identity during the edit? Additionally, are there specific conditions or types of text prompts that particularly enhance this preservation aspect within the CLIPDrag framework?

---

> ### Author Response · Authors · 2024-11-24
> **Response to Reviewer CxEc(1/2)**
>
> Thanks for the constructive comments. We try to address your concerns  point by point as follows:
>
> > W1:Novelty of FPT: The paper should acknowledge that searching for handle points along the path from handle points to targets has been previously explored in methods like DragGAN and DragDiffusion. To clarify the unique contributions of FPT, the authors should provide side-by-side comparisons of point tracking strategies, highlighting any improvements or distinctions in their approach.
>
> The principle underlying all point tracking methods is consistent: they use a nearest-neighbor search algorithm to update the positions of handle points. The key distinction between our FPT strategy and the approach used in DragGAN/DragDiffusion lies in the search area, as illustrated in Figure 3(c).
>
> Specifically, DragGAN/DragDiffusion searches for new handle points within a square patch centered around the current handle positions:
>
> $$h^{k+1} _ i = \arg\min _ {a\in\Omega(h^k _ i,r _ 2)} || F_ q(z^{k+1} _ t)-F _ {h^k _ i}(z^0 _ t)|| $$
>
> However, this approach has notable limitations, as shown in Figure 8. Handle points may move in the wrong direction or even form a loop, leading to inefficient optimization and requiring more iterations to complete the drag edit.
>
> To address this issue, we introduced a simple constraint on the search area: $dist(a, g _ i)< dist(h^k_i,g _ i)$. This ensures that our FPT method only considers candidate points that are closer to the target points, effectively transforming the optimization into a monotonic process:
>
> $$h^{k+1} _ i = \arg\min _ {a\in\Omega(h^k _ i,r _ 2) \And dist(a,g _ i)< dist(h^k _ i,g _ i)} || F_q(z^{k+1} _ t)-F _ {h^k _ i}(z^0 _ t)|| $$
>
> As demonstrated in Figure 8(d), our FPT strategy achieves similar editing results with significantly fewer iterations, thereby accelerating the editing process.
>
>
>
> >W2: More extensive comparisons or discussions with recent techniques such as InstantDrag, LightningDrag, StableDrag, and RegionDrag. Additionally, these methods should be thoroughly discussed in the related work section to provide a more complete context.
>
> Thanks for the constructive suggestions. To better demonstrate CLIPDrag's ability to address the ambiguity phenomenon, we give the result of InstantDrag, LightningDrag, StableDrag, and RegionDrag in our appendix(Appendix A).  As we can see， all these methods have the problem of ambiguity(InstantDrag, LightningDrag) or identity-preserving(StableDrag, RegionDrag).
>
>
>
> Also, we revise the content of the related work and discuss these methods in detail.
>
>
> >W3: Performance Metrics. The paper should include a discussion or report on inference time comparisons.
>
> We understand the concern and report the average inference time of CLIPDrag, DragDiff, SDE-Drag and FreeDrag as follows:(the result is calculated on a single 3090 GPU by averaging over 100 examples sampled from the DragBench.) This part is also included in Appendix F.
>
> | Method         | Inference Time |
> | -------------- | -------------- |
> | DragDiff       | 80.3s          |
> | FreeDrag       | 69.4s          |
> | FastDrag       | 75.5s          |
> | SDE-Drag       | 70.0s          |
> | StableDrag     | 72.3s          |
> | CLIPDrag(Ours) | 47.8s          |
>
> As shown in the table, our method is significantly faster than previous works. This is because the inference time is directly correlated with the number of optimization iterations. In CLIPDrag, the text guidance helps to indicate the correct optimization direction, while our FPT strategy prevents handles from moving in the wrong direction or forming loops. Both of these factors reduce the number of iterations required to move the handle points to their target positions, resulting in faster editing speeds.

---

> > ### Comment · Reviewer_CxEc · 2024-11-25
> >
> > Thank you for your detailed rebuttal. Your clarification of the experimental methodology and additional data analysis effectively addresses my initial concerns.
> >
> > One remaining concern is about the evaluation of FastDrag's inference time, as they report an inference time of less than 5 seconds.

---

> > > ### Author Response · Authors · 2024-11-25
> > > **Follow-up from reviewer CxEc.**
> > >
> > > Thanks for your concern. This is because the setting in our paper is: **Making sure all handle points can reach target points** (lines 320,321).  So we have a larger **maximum iteration number** (2000), while in previous methods like DragDiff/StableDrag,  this hyper-parameter is set to 80, which means the optimization will stop even if handle points are far away from the target points.
> > > So for fairness, although FastDrag is a one-step editing method, it needs to run its algorithm many times if the handle points are checked to be far away from the target points, which significantly increases the edit time.
> > >
> > > We hope this answers your questions. Thank you again for your valuable feedback, and please don’t hesitate to let us know if there are follow-up questions.

---

> > > > ### Comment · Reviewer_CxEc · 2024-11-25
> > > >
> > > > Thank you for the reply, all my concerns have been fully addressed.

---

> ### Author Response · Authors · 2024-11-24
> **Response to Reviewer CxEc(2/2)**
>
> >w4: User Input Optimization. The authors could explore incorporating vision-language models like GPT-4V to automatically interpret the input image (as shown in the first column of Figure 4). This approach could significantly reduce user burden while maintaining the benefits of text-guided editing.
>
> Thank you for your valuable suggestions. We agree that incorporating vision-language models, such as GPT-4V, to automatically generate prompts would make our method more user-friendly. Following your advice, we plan to implement the following modifications in our code to integrate a large vision-language model:
>
> 1. If a text input is detected, it will be used for fine-tuning and editing.
> 2. If no text prompt is detected, the GPT-4V API will be called to generate a caption for the input image and the resulting caption will be used for subsequent operations.
>
> This modification is included in the revised manuscript.(line 269)
>
> >Q1: Could you elaborate on how the addition of text signals specifically contributes to preserving the object's original identity during the edit? Additionally, are there specific conditions or types of text prompts that particularly enhance this preservation aspect within the CLIPDrag framework?
>
> **How does the addition of text signals specifically contribute to preserving the object's original identity during the edit?**
>
> Text signals contribute to preserving the object's original identity by providing gradient information that guides both local edits and global preservation. While drag-based image editing focuses primarily on aligning local features, text-based editing inherently considers both region-specific modifications and the overall preservation of identity. The gradient of the text signals  $G_g$ thus contains two components: one for editing specific regions and another for preserving global identity.
>
> If the edit component is known, we can extract the identity-preservation information through decomposition (as shown in Figure 3(a)(b)). Since drag signals are focused on editing local regions, the direction perpendicular to them represents the identity-preserving direction. Consequently, as shown in Equation (6), when the edit direction of the text signals and drag signals align, the identity-preserving component of the text gradient is added to maintain global features.
>
> **Are there specific conditions or types of text prompts that particularly enhance this preservation aspect within the CLIPDrag framework?**
>
> Yes, when the text prompt is simply a description of the object, we found that the gradient of the text signals and the drag signals are nearly orthogonal (i.e., $\sin <G_g, G_l>$ close to 1 in Equation 6). This means that almost all of the gradient information from the text signals is used for preserving the global identity of the object, rather than altering local features.
>
> **References**
>
> [1]Shin, Joonghyuk, Daehyeon Choi, and Jaesik Park. "InstantDrag: Improving Interactivity in Drag-based Image Editing." arXiv preprint arXiv:2409.08857 (2024).
>
> [2]Shi, Yujun, et al. "LightningDrag: Lightning Fast and Accurate Drag-based Image Editing Emerging from Videos." arXiv preprint arXiv:2405.13722 (2024).
>
> [3]Cui, Yutao, et al. "StableDrag: Stable Dragging for Point-based Image Editing." ECCV (2024).
>
> [4]Lu, Jingyi, Xinghui Li, and Kai Han. "RegionDrag: Fast Region-Based Image Editing with Diffusion Models." ECCV (2024).
>
> [5]Zhao, Xuanjia, et al. "FastDrag: Manipulate Anything in One Step." NeurIPS (2024).
>
> [6]Nie, Shen, et al. "The blessing of randomness: Sde beats ode in general diffusion-based image editing."ICLR (2024).

---

### Official Review · Reviewer_qFtD · 2024-10-31

**Soundness:** 3
**Presentation:** 2
**Contribution:** 3
**Rating:** 5
**Confidence:** 3

**Summary:**

CLIPDrag combines text and drag-based controls to improve image editing, using text for broad guidance and drag points for precise adjustments. The author introduces Global-Local Motion Supervision, which combines gradients from both text and drag inputs, and Fast Point Tracking to speed up convergence. This method eliminates common issues like vagueness in text-only edits and ambiguity in drag-only edits.

**Strengths:**

1.	The motivation is clear and effective, combining text and drag editing to leverage the strengths of both approaches, achieving more precise edits.
2.	The Global-Local Gradient Fusion method is innovative, merging global text and local drag gradients to enhance editing quality, with experiments showing notable improvements in performance.

**Weaknesses:**

1.	The illustration in Figure 2 is unclear in terms of workflow. If CLIP guidance is applied, the latent space should ideally be converted to the pixel domain to align with CLIP’s processing. However, the diagram uses SD rather than a VAE.
2.	CLIPDrag lacks comprehensive quantitative comparisons with other methods in image editing. The current evaluation only includes DragDiff in Figure 6, which is insufficient.
3.	The ablation study also lacks more detailed quantitative comparisons. In Figure 8, the visual differences between (b) and (c) are subtle, making it hard to discern the impact of changes.

**Questions:**

The comparisons in this paper in insufficient, why only DragDiff is compared to in this paper? More comparisons should be added.

---

> ### Author Response · Authors · 2024-11-24
> **Response to Reviewer qFtD(1/2)**
>
> We sincerely appreciate your constructive comments to improve our paper and detail our response below.
>
> > The illustration in Figure 2 is unclear in terms of workflow. If CLIP guidance is applied, the latent space should ideally be converted to the pixel domain to align with CLIP’s processing. However, the diagram uses SD rather than a VAE.
>
> We adopt the method from the DDIM paper to convert the latent $z_t$ into the corresponding image: the latent $z_t$ is first input into the diffusion part $\epsilon_\theta$ of the SD to obtain the predicted noise $\epsilon_\theta(z_t,t)$ and then get the predicted initial latent $\hat{z_0}$ :
>  $$\hat{z_0} = \frac{z_t - \sqrt{1-\tilde{\alpha_t}}\cdot\epsilon_\theta(z_t)}{\sqrt{\tilde{\alpha_t}}}$$
>
> Next, $\hat{z_t}$ is converted to the pixel domain by the VAE part of SD. As you suggested, the leftmost part of Figure 2 should indeed represent the VAE, as the DDIM inversion does not involve the diffusion process. However, the subsequent projections should be attributed to SD, as both the diffusion part and the VAE are involved.
>
> We recognize that this might have been misleading, and to clarify this, we have added a footnote in the manuscript.(line 215)
>
>
> > W2: CLIPDrag lacks comprehensive quantitative comparisons with other methods in image editing. The current evaluation only includes DragDiff in Figure 6, which is insufficient.
>
> To achieve a more comprehensive comparison of the drag edit experiment in Figure 6, we have added the results of two recent drag-based methods, InstantDrag and FastDrag.
> The corresponding visual results are as in Appendix E.
>
> Besides, here is the quantitative analysis on DragBench:
>
> **Mean Distance(MD), the lower the better**
>
> NOTE: the number(10,20,40,80,160) means the maximum iterations,
>
> | Method     | 10   | 20   | 40   | 80   | 160  |
> | ---------- | ---- | ---- | ---- | ---- | ---- |
> | Ours       | 49.5 | 45.1 | 39.2 | 35.8 | 32.3 |
> | DragDiff   | 51.3 | 50.6 | 42.9 | 38.8 | 35.1 |
> | StableDrag | 50.8 | 48.8 | 42.3 | 39.0 | 34.8 |
> | FastDrag   | 52.1 | 51.1 | 44.4 | 41.9 | 36.9 |
>
>
>
> **Image Fedlity, the higher, the better**
>
> | Method     | 10   | 20   | 40   | 80   | 160  |
> | ---------- | ---- | ---- | ---- | ---- | ---- |
> | Ours       | 0.95 | 0.94 | 0.93 | 0.90 | 0.88 |
> | DragDiff   | 0.95 | 0.93 | 0.90 | 0.87 | 0.85 |
> | StableDrag | 0.95 | 0.90 | 0.88 | 0.84 | 0.81 |
> | FastDrag   | 0.95 | 0.93 | 0.93 | 0.89 | 0.83 |
>
>
>
> We observe that compared to DragDiff, StableDrag achieves more stable image edits, meaning the object's identity is better preserved. However, it may not always perfectly align the handle features with the target points. In contrast, FastDrag tends to perform the opposite, where the handle features are more accurately dragged to the target positions, but the stability of the image edit (and preservation of identity) is somewhat compromised.

---

> ### Author Response · Authors · 2024-11-24
> **Response to Reviewer qFtD(2/2)**
>
> >W3:The ablation study also lacks more detailed quantitative comparisons. In Figure 8, the visual differences between (b) and (c) are subtle, making it hard to discern the impact of changes.
>
> We would like to provide some clarification regarding Figure 8. In this ablation study, we compare our FPT strategy with the point tracking (PT) strategy used in DragGAN/DragDiff.  Because the primary goal of FPT is to accelerate the editing process,  to ensure a fair comparison we have to make sure the visual differences between panels (b) and (c) are subtle, which means FPT achieves similar editing results as PT but at a faster pace. Therefore we can conclude that FPT can speed up the edit process without compromising the quality of the edits.
>
>
>
>
>
> > Q1: The comparisons in this paper is insufficient, why only DragDiff is compared  in this paper? More comparisons should be added.
>
> Thanks for the suggestion. To better show the performance and generalization of our CLIPDrag method, we have added the result of other drag-based methods (SDE-Drag, InstantDrag, RegionDrag, and LightningDrag), shown in Appendix A.  As we can see， all these methods have the problem of ambiguity(InstantDrag, LightningDrag) or identity-preserving(StableDrag, RegionDrag).
>
> **References**
>
> [1]Shin, Joonghyuk, Daehyeon Choi, and Jaesik Park. "InstantDrag: Improving Interactivity in Drag-based Image Editing." arXiv preprint arXiv:2409.08857 (2024).
>
> [2]Shi, Yujun, et al. "LightningDrag: Lightning Fast and Accurate Drag-based Image Editing Emerging from Videos." arXiv preprint arXiv:2405.13722 (2024).
>
> [3]Cui, Yutao, et al. "StableDrag: Stable Dragging for Point-based Image Editing." ECCV (2024).
>
> [4]Lu, Jingyi, Xinghui Li, and Kai Han. "RegionDrag: Fast Region-Based Image Editing with Diffusion Models." ECCV (2024).
>
> [5]Zhao, Xuanjia, et al. "FastDrag: Manipulate Anything in One Step." NeurIPS (2024).

---

> ### Author Response · Authors · 2024-11-27
> **Follow-up**
>
> Dear Reviewer,
>
> As the deadline for the author-reviewer discussion phase is approaching, we would like to check if our response addressed your concerns. If there are any remaining issues or if you require further clarification, please feel free to inform us.
>
> Thanks!

---

> ### Author Response · Authors · 2024-12-02
> **Official Comment by Authors**
>
> Dear Reviewer qFtD,
>
> **As the rebuttal discussion period ends in two days**, we would be grateful for your feedback on whether our responses have adequately addressed your concerns. We are ready to answer any further questions you may have.
>
> Thank you for your valuable time and effort!
>
> Best regards,
>
> The Authors

---

### Official Review · Reviewer_R2Bm · 2024-11-04

**Soundness:** 2
**Presentation:** 2
**Contribution:** 2
**Rating:** 3
**Confidence:** 5

**Summary:**

This paper proposes a Text-Drag Editing framework to address text-based and drag-based editing limitations. To achieve this, the authors introduce global-local motion supervision that integrates the semantic aspects of text with drag guidance. They utilize a novel approach of gradient fusion, combining gradients from text and drag conditioning based on their directional alignment to provide a unified gradient.

**Strengths:**

For the first time, the paper provides an algorithm that integrates text-guided editing with drag-guided editing. The proposed editing algorithm attempts to provide more precise global editing and reduce ambiguity in local editing. The independent guidance or supervision of text and drag is combined interestingly by disentangling the global gradient that is perpendicular and parallel to the local gradient.

**Weaknesses:**

1. Lack of Comprehensive Review of Diffusion-Based Image Editing Literature:
The paper does not provide an adequate overview of diffusion-based image editing methods. A more thorough review of recent approaches in diffusion-based image editing is necessary to strengthen its background and situate the proposed method within the broader field. Specifically, the authors should consider discussing recent methods, such as
SINE: SINgle Image Editing With Text-to-Image Diffusion Models (Zhang et al., CVPR 2023),
Paint by Example: Exemplar-based Image Editing with Diffusion Models (Yang et al., CVPR 2023),
FlexiEdit: Frequency-Aware Latent Refinement for Enhanced Non-Rigid Editing (Koo et al., ECCV 2024),
and RegionDrag: Fast Region-Based Image Editing with Diffusion Models (Lu et al., ECCV 2024).
Incorporating these examples will provide a more robust foundation and context for the reader, enabling a clearer understanding of how the current approach builds upon or diverges from existing work.

2. Unconvincing Example in Figure 1:
The example provided in Figure 1 does not convincingly illustrate the motivation of the study. The intention seems to highlight the limitations of drag-based and text-based editing approaches, yet the figure only demonstrates an instance where drag-based editing is ineffective. A more persuasive example might involve a scenario where drag-based editing produces a subtle change—such as adjusting a subject's smile—which could then be further refined by the proposed text-drag editing method to achieve a more detailed, natural effect. This change would clarify the benefits of text-drag editing over existing methods.

Additionally, the similarity between the proposed method's results and traditional drag-based editing in Figure 1 and the statute example raises questions about the added benefit of the proposed approach. If these similarities are not intentional, a different example or refinement of the illustrations might better demonstrate the unique advantages of the proposed method.

3. Handling of Distinct Effect Regions in Text-Based and Drag-Based Editing
The paper does not adequately explain how it manages distinct effect regions associated with text-based and drag-based editing despite these methods likely targeting different areas in an image. Clarifying how these regions are defined, integrated, or adjusted during editing would provide more specificity and improve understanding of the algorithm's functionality. This discussion is crucial to distinguish the contribution of the combined editing approach.

4. Suggested Comparative Experiments for Method Validation
Comparative experiments should include scenarios where text-based editing is applied after drag-based editing and vice versa to illustrate the proposed method's effectiveness better. This comparison would help demonstrate the practical advantage of combining both methods in the proposed approach and establish whether there are meaningful improvements when they are applied sequentially.

5. Limited Novelty in Gradient Combination Approach
The novelty presented in Equation (6), which combines the two editing approaches by decomposing one gradient into the other perpendicular component and then summing them, seems linear, and it is conceivable that a non-linear combination may provide a more effective result.  Including alternative approaches as comparative experiments would strengthen the paper's case for its approach or help contextualize its performance relative to existing methods.

The paper introduces a combined text-drag editing approach but lacks a comprehensive literature review, convincing examples, clarity regarding region specificity, and evidence of sufficient novelty. Addressing these areas would help elevate the study’s contributions and clarify its position within diffusion-based image editing.

**Questions:**

Please provide examples with and without drag edit the following examples. "The sculpture is smiling and not showing his teeth." and "The sculpture is smiling and not raising his head".

---

> ### Author Response · Authors · 2024-11-24
> **Response to Reviewer R2Bm(1/2）**
>
> We sincerely appreciate your constructive comments on improving our paper. We detail our response below and have corrected the corresponding part in our revision.
>
> >W1: Lack of Comprehensive REview of Diffusion-Based  Image Editing.
>
> Thank you for the suggestions. We have updated the related work section in our latest manuscript version. Specifically, we have made the following improvements:
>
> 1. A subsection was added to introduce the development of diffusion models(lines 128-141).
> 2. Included recent methods in text-based image editing, such as SINE, Paint by Example, and FlexEditlines 147-150).
> 3. Provided a more detailed overview of drag-based methods, including RegionDrag, FastDrag, FreeDrag, InstantDrag, and other recent approaches(lines 160-200).
>
> >W2: Unconvincing Example in Figure 1.
>
> Thanks for the suggestion. To better show our motivation, We have made some modifications in Figure 1. Since the first example has already demonstrated the function of text signals, for the second case of the sculpture, we try to show the effect of drag signals. Specifically, we keep the text prompt "The sculpture is smiling" unchanged, by changing the position of drag points, CLIPDrag can control the extent of the smile: showing or not showing the teeth.  We hope this modification will help readers understand our motivation.
>
>
> >W3: Handling of Distinct Effect Regions in Text-Based and Drag-Based Editing.
>
> **How are effect regions defined?**
>
> The effect regions in drag-based editing are clearly defined by the patch features of handle points and target points. However, the effect regions of text signals are not explicitly defined. Instead, they correspond to positions with high attention scores in the cross-attention map of the U-Net. For instance, in the text prompt "A photo of a smiling woman," the effect regions would include the woman's mouth (related to 'smiling') and other areas like the face or eyes (related to 'woman').
>
> In our method, we categorize all effect regions into two types:
> (i) Edit regions: These are the areas where we aim to change features, such as regions around the drag points and tokens like "smiling."
> (ii) Identity regions: These are the areas where we want to preserve features, such as the regions corresponding to "woman."
>
>
> **How effect regions are integrated or adjusted?**
>
>
> Actually, we integrated these effect regions through gradient fusion, but the Gradient Fusion can be explained from the perspective of regions. The effect regions of the edit component of $G_g$ align with the edit regions, and similarly for the identity component.
>
> When the edit regions of the text signal are consistent with the drag points' effect region, we use the drag signals to modify the image and the text signals to preserve identity, as shown in Figure 3(a). When the edit regions of the text signals contradict the drag points' effect regions, we use the edit regions of the text signals to adjust the effect regions of the drag signals. The integration and adjustment of these regions are achieved through gradient fusion.

---

> ### Author Response · Authors · 2024-11-24
> **Response to Reviewer R2Bm(2/2)**
>
> >W4: Suggested Comparative Experiments for Method Validation.
>
> We understand the reviewer’s concern and would like to clarify why we did not consider applying the two types of guidance sequentially or in the opposite order.
>
> **Why not text-based edit after drag-based edit?**
>
> The goal of our paper is to perform drag-based editing while eliminating ambiguity with the help of a text prompt. If we were to apply drag-based editing first, the optimization direction of the latent could be incorrect, meaning that the ambiguity problem would occur before the text guidance is applied.
>
> For example, consider Figure 4(d): If the drag signal is applied first, the model would edit the image towards the direction of "Enlarging the woman's face." Consequently, the prompt "make the woman smile" would modify the image based on an incorrect initial edit, leading to results that are not consistent with the user’s intention.
>
> Additionally, if text-based editing were applied after the drag operation, the position of the target points would be altered. As a result, after the text-based edit, the final positions of the handles would no longer align with the targets, which contradicts the core principle of drag-based methods.
>
>
> **Why not drag-based edit after text-based edit?**
>
>
>  This approach would also introduce ambiguity. When text-based editing is applied first, the position of the handle points is altered. This alteration can mislead the subsequent drag operation. For example, in Figure 4(d), after applying the text guidance "make the woman smile", the new handle points might end up at the top right of the target points. As a result, when the drag operation tries to move the handles to the targets, it could imply a semantic change like "Make the woman not smile," which contradicts the original intent.
>
> In addition to this analysis, we also give some results in Appendix C when the two signals are applied sequentially.
>
> >W5: Limited Novelty in Gradient Combination Approach.
>
>
> We completely agree that a non-linear combination of local and global gradients could improve performance, and designing better gradient fusion strategies is a promising direction for future work. In fact, our gradient fusion strategy is implemented as a plug-and-play method in the official code, making it easy to experiment with different fusion approaches.
>
> However, we would like to emphasize that the main focus of our work is to address the ambiguity problem in drag-based image editing. Since no prior work has explored incorporating text signals in this context, our primary contribution is to propose a novel idea for combining these two signals from the perspective of gradients. The gradient combination approach (GLGF) was introduced to demonstrate that gradients can serve as an effective medium for merging the two signals. This is why we chose not to delve into the specifics of gradient fusion design in this paper.
>
> >Q1:provide the examples.
>
> Yes, according to your suggestion, we have added four results with and without drag edit using the prompt: "The sculpture is smiling and not showing his teeth." and "The sculpture is smiling and not raising his head". These results are included in Appendix B.

---

> ### Author Response · Authors · 2024-11-27
> **Follow-up**
>
> Dear Reviewer,
>
> As the deadline for the author-reviewer discussion phase is approaching, we would like to check if our response addressed your concerns. If there are any remaining issues or if you require further clarification, please feel free to inform us.
>
> Thanks!

---

> ### Author Response · Authors · 2024-12-02
> **Official Comment by Authors**
>
> Dear Reviewer R2Bm,
>
> **As the rebuttal discussion period ends in two days**, we would be grateful for your feedback on whether our responses have adequately addressed your concerns. We are ready to answer any further questions you may have.
>
> Thank you for your valuable time and effort!
>
> Best regards,
>
> The Authors

---

### Official Review · Reviewer_RgXk · 2024-11-09

**Soundness:** 3
**Presentation:** 4
**Contribution:** 3
**Rating:** 8
**Confidence:** 4

**Summary:**

This paper introduces CLIPDrag, a novel image editing approach that integrates both text-based and drag-based controls to achieve more precise and flexible edits. Traditional text-based editing provides general guidance but often lacks specificity, while drag-based editing offers local control but can be ambiguous without context. CLIPDrag addresses these issues by using text as global guidance for overall image context and drag points for fine-grained, localized control. The model leverages a global-local motion supervision (GLMS) system and a fast point-tracking (FPT) method to streamline and accelerate the editing process. The paper is well written and easy to understand, the paper has comprehensive experimental results which show CLIPDrag outperforms both traditional drag- and text-based methods in accuracy and image fidelity. The detailed ablations make the hypothesis clear. The paper presents an interesting path for image editing and is theoretically grounded which should be shared within the community

**Strengths:**

1. Novel Approach to combine local and global gradient: Building on text inversion methods to combine text and drag signals, CLIPDrag enables pixel-level control, offering both specific and contextually aware edits.
2. Efficient Convergence: The fast point-tracking method improves the editing process by guiding handle points toward their target positions faster.
3. Extensive Ablations: The paper has ablations for all different components such as point tracking, GLMS and controls with edit and text showing clear performance gain.
4. Qualitative Results: The papers presents representative set of results allowing easy intuition and help with clarity of the paper.

**Weaknesses:**

1. The need for identity preservation beyond citing DragDiffusion is not shared, given the improvement of base models, the intuition behind it is lacking.
2. Gradient accumulation is discussed assuming the latent code is continuous, formulating why the gradient manipulation will still lead to plausible images is unclear.
3. Assumption around using nearest neighbors in FPT moving monotonically towards target is not explained, given the optimization is highly non linear.

**Questions:**

1. What are some common failure cases for the editing, especially if the text and local edits conflict.
2. How are the number of iterations for denoising fixed for drag editing and how do the impact change with fewer to larger iterations.
3. One of example is shown to incorporate masks for editing, can it be explained how masks are incorporated in this framework ?

---

> ### Author Response · Authors · 2024-11-24
> **Response to Reviewer RgXk(1/2）**
>
> We thank the reviewer for acknowledging the novelty of the proposed method the promising experimental results, and the organization of the paper presentation. We will respond to your concerns one by one as follows:
>
> > W1:  The need for identity preservation beyond citing DragDiffusion is not shared.
>
> We appreciate the reviewer’s feedback and agree that providing the intuition behind the identity preservation step will enhance the clarity of the paper.
>
> Identity preservation is crucial because drag-based methods primarily focus on regional features during image editing—i.e., they optimize latent variables based on differences between localized feature patches. This localized focus can inadvertently compromise global information, such as the object’s structure. By fine-tuning a pre-trained diffusion model in advance, identity preservation helps the model maintain the image’s overall structural integrity throughout the editing process.
>
> This rationale is further supported by the ablation study by DragDiffusion. The study demonstrates that while image editing without identity preservation can achieve semantic modifications, it often alters global attributes such as the background. Consequently, identity preservation has become a standard step in drag-based methods, including the one proposed in this paper.
>
>
> > W2:  why the gradient manipulation will still lead to plausible images.
>
> Thank you for the question. The key reason for this behavior is the robustness of diffusion models to noise. The result of gradient manipulation is the modification of the optimized latent. Each gradient manipulation corresponds to a small perturbation of the original latent, and these perturbations accumulate across iterations.
>
> When the handle points reach the target points within just a few iterations, the perturbation remains minor, allowing the denoising process to effectively handle it and produce a plausible image. However, if the handles take more iterations to reach the targets, the accumulated perturbation becomes larger, which may exceed the diffusion model's ability to handle it, potentially affecting image quality.
>
> > W3:  why the FPT strategy can make handles move monotonically towards targets.
>
> Both our FPT strategy and DragDiff’s point tracking (PT) mechanism use a nearest-neighbor search to update the positions of handles. However, as you noted, this approach does not inherently ensure that handle points will move monotonically toward target points. This limitation arises because the naïve point tracking method searches for new handles within a square patch centered on the current handle positions, as defined by:
>
> $$ h^{k+1} _ i = \arg\min _ {a\in\Omega(h^k _ i,r _ 2)} || F_q(z^{k+1}  _ t)-F_{h^k _ i}(z^0 _ t)|| $$
>
> To address this issue, we introduced a simple constraint on the search patch: $dist(a,g_i)< dist(h^k_i,g_i)$. This constraint ensures that our FPT method only considers candidate points that are closer to the target points, effectively converting the optimization into a monotonic process:
>
> $$ h^{k+1} _ i = \arg\min _ {a\in\Omega(h^k _ I, r _ 2) \And dist(a,g _ i)< dist(h^k _ i,g _ i)} || F_q(z^{k+1} _ t)-F_{h^k _ i}(z^0 _ t)|| $$
>
> With this adjustment, the new handles move closer to the target points after each iteration, as candidates farther from the targets are excluded from consideration.

---

> ### Author Response · Authors · 2024-11-24
> **Response to Reviewer RgXk(2/2）**
>
> > Q1: what are some failure cases when the text and local edit conflicts?
>
> Thank you for raising this concern. While our primary motivation is to use the text prompt to complement the local drag edit—ensuring that these two signals are consistent in most cases—we also explored scenarios where these guidance signals conflict. Based on our experiments, we observed two types of potential editing outcomes:
>
> 1. Ambiguity or Neutralization at Moderate Text Strength ($\lambda$<10). When the strength of the text signal is moderate, ambiguity can arise if the text guidance fails to provide accurate information. In other cases, the text signal may counteract the drag operation, effectively neutralizing its effect. For instance, in Figure 4(e), the drag instruction combined with the prompt "Make the heel of the shoes higher" might yield a result akin to "Make the heel not so high."
>
> 2. Implausible Results at High Text Strength($\lambda=100$). When the text guidance is excessively strong, it overwhelms the denoising process, making it difficult to handle the perturbation. This can result in implausible or unrealistic edits.
>
> You can see the corresponding example in Appendix D.
>
>
>
>
>
> > Q2: How are the number of iterations for denoising fixed for drag editing and how does the impact change with fewer to larger iterations?
>
>
> Thank you for your questions. Below, we address them individually:
>
>
> **How are the (max) number of iterations fixed?**
>
> The number of iterations is a hyperparameter in drag-based methods, representing the maximum number of optimization steps allowed during editing. If the handle points fail to reach the target positions within this limit, the model halts latent optimization and directly denoises the latent to generate the final image. In previous methods, this value was typically set to 80; however, in our work, we increased it to 2000 to explore the effects more comprehensively.
>
> **How does the impact change with fewer to larger iterations?**
>
> When the maximum number of iterations is larger, handle points are more likely to reach the target positions. However, this comes at the potential cost of image quality due to error accumulation during the denoising process.
>
> It is about the error accumulation in the denoising process. During each iteration, drag methods optimize the latent to move the features of handle points to target points slightly (Motion Supervision) and then relocate the position of new handle points in the feature map of the optimized latent(Point Tracking). This iterative process can be thought of as applying small perturbations to the latent representation, and as the iterations increase, the perturbation accumulates. Therefore, if the handles reach targets within small iterations, the perturbation is minor and can be handled by the denoising process due to the robustness of the diffusion model. Consequently, the edited image is plausible and the semantics are changed precisely.
>
> However, in many cases(like the example of Fig. 8), the handles may move in the wrong direction or even form a loop, which means more iterations are needed. Then there is a trade-off between the semantic revision and image fidelity:
>
> 1. Fewer iterations: Handle points may fail to reach the targets, leading to incomplete semantic revisions.
> 2. Larger iterations: While achieving better alignment between handle and target points, the excessive perturbation can compromise image fidelity.
>
>
>
> > Q3: One of example is shown to incorporate masks for editing, can it be explained how masks are incorporated in this framework?
>
> Thanks for the suggestion. The mask is used to specify an editable region to achieve the desired editing and is specified by the user. If a mask is given, a related term is added to $L_{ms}$ in Equation (5)  :
> $$||(z^k_t - sg(z^0_t)) \odot (\mathbb{1}-M)||_1$$
> where, $z^k_t$, $z^0_t$ are the t-th and 0-th step latent, $M$ is the corresponding mask. As we can see, this term encourages the unmasked area to remain unchanged in the motion supervision phase. (line 284)

---

### Author Response · Authors · 2024-11-24
**General Response to All Reviewers.**

We appreciate their suggestions and comments and carefully revise our paper accordingly. Our major revisions include the following four aspects:

1. In the Introduction, we revised the second example in Figure 1 for better illustration.
2. In the Related Work, we included the discussion of diffusion model(lines 128-141) and provided a more comprehensive discussion of drag-based methods(lines 160-200).
3. In the Experiments:
   -   We added four more baselines in the drag-text edit setting(line 355).
   -   We added explanations of the pipeline diagram(line 214,269).
   -   We compared two more methods in the quantitative experiment of drag-based editing(Figure 6).

4. In the Appendix:
   - Appendix A shows more drag-based results on text-drag edit.
   - Appendix B shows four more examples to explain the motivation.
   - Appendix C shows results when applying text and drag guidance sequentially.
   - Appendix D shows  examples when text and drag signals conflict
   - Appendix E shows examples of StableDrag, and FreeDrag on drag-based editing.
   - Appendix F shows the inference time comparisons.

Please note that we colorized (blue) the revisions in the new version of the paper.

---

### Meta-Review · Area_Chair_KA2d · 2024-12-21

**Metareview:**

This paper proposes an image editing method called CLIPDrag, which aims to combine text and drag signals for unambiguous manipulations on diffusion models.

This paper received mixed initial reviews with ratings of 8, 6, 5, and 3. After the rebuttal, one reviewer increased the score from 6 to 8, while others maintained their initial ratings. The final ratings remained varied at 8, 8, 5, and 3. It should be noted that the reviewers who gave negative ratings of 3 and 5 were unresponsive during the rebuttal and discussion phases. The area chair had encouraged their involvement but did not get further feedback. As a result, the area chair considered lowering the weight of their reviews.

Despite the unresponsiveness of those two reviewers, the area chair felt that most of the concerns raised were adequately addressed by the authors, for example,
* Adding subsections to review diffusion models, text-based image editing methods, and drag-based methods;
* Modifying Figure 1 to illustrate the idea of Text-Drag Edit;
* Explaining how are effect regions defined and integrated;
* Clarifying why not considering applying the two types of guidance sequentially.

On the other hand, the reviewer who increased the rating from 6 to 8 acknowledged the detailed rebuttal from the authors, stating that  *"[Y]our clarification of the experimental methodology and additional data analysis effectively addresses my initial concerns."* After further discussion, the reviewer's additional concern regarding the inference time was also fully addressed.

Given that two reviewers expressed definitive support for the paper, the area chair concurs with their suggestions and recommends accepting the paper.

**Additional Comments On Reviewer Discussion:**

Reviewers pointed out concerns about the FPT strategy, such as
* Why the FPT strategy can make handles move monotonically towards targets?
* To clarify the unique contributions of FPT, the authors should provide side-by-side comparisons of point searching strategies, highlighting any improvements or distinctions in their approach.

The authors are encouraged to include the discussions in Sec. 3.3 of the paper.

---

### Decision · Program_Chairs · 2025-01-22

Accept (Poster)